# 3-D Data Interpolation and Denoising by an Adaptive Weighting Rank-Reduction Method Using Multichannel Singular Spectrum Analysis Algorithm

**DOI:** 10.3390/s23020577

**Published:** 2023-01-04

**Authors:** Farzaneh Bayati, Daniel Trad

**Affiliations:** Department of Earth Science, University of Calgary, 2500 University Drive NW, Calgary, AB T2N 1N4, Canada

**Keywords:** seismic data, interpolation, MSSA, SVD, rank reduction

## Abstract

Addressing insufficient and irregular sampling is a difficult challenge in seismic processing and imaging. Recently, rank reduction methods have become popular in seismic processing algorithms for simultaneous denoising and interpolating. These methods are based on rank reduction of the trajectory matrices using truncated singular value decomposition (TSVD). Estimation of the ranks of these trajectory matrices depends on the number of plane waves in the processing window; however, for the more complicated data, the rank reduction method may fail or give poor results. In this paper, we propose an adaptive weighted rank reduction (AWRR) method that selects the optimum rank in each window automatically. The method finds the maximum ratio of the energy between two singular values. The AWRR method selects a large rank for the highly curved complex events, which leads to remaining residual errors. To overcome the residual errors, a weighting operator on the selected singular values minimizes the effect of noise projection on the signal projection. We tested the efficiency of the proposed method by applying it to both synthetic and real seismic data.

## 1. Introduction

Seismic surveys are designed to keep a consistent grid of sources and receivers. In typical surveys, regularly sampled seismic surveys are uncommon or rare because of logistic obstacles or economic constraints. These limitations lead to large shot and receiver sampling intervals, which produce poorly, irregularly sampled seismic data along spatial coordinates with gaps without recorded traces. Seismic reconstruction methods can be divided into four main classes: signal processing-based methods, wave equation-based methods, machine learning-based methods, and rank reduction-based methods.

Most of the methods in the signal processing-based category are multidimensional and use prediction filters [1,2,3] and transform domains through means such as the Fourier transform [4,5,6], Radon transform [7,8], or Curvelet transform [9]. There are some hybrid techniques using combinations of the Fourier transform with prediction error filters [10] as a way to improve interpolation beyond aliasing. Vassallo et al. [11] proposed multichannel interpolation by matching pursuit (MIMAP), which interpolates high-order aliased data without making any prior information about the linearity of seismic events or the seismic wavefield model. Ghaderpour [12] proposed a multichannel anti-leakage least-squares spectral (MALLSSA) method that includes the spatial gradients of seismic data into the anti-leakage least-squares spectral (ALLSSA) to regularize seismic data beyond aliasing.

Wave-equation-based algorithms execute an implicit migration de-migration pair. Stolt [13] introduced a reconstruction algorithm that regularized a dataset by mapping it to physical space using migration–demigration. A finite-difference offset continuation filter was proposed by Fomel [14].

In recent years, machine learning has attracted much attention in geophysical studies. Deep learning (DL) is having a great influence on signal and image processing. However, although very powerful, machine learning for geophysical problems is not mature yet, and applications often ignore the complications and subtleties of real data processing. Wang et al. [15] proposed a ResNet block design to interpolate anti-aliased seismic data. Siahkoohi et al. [16] proposed generative adversarial networks (GANs) to reconstruct heavily sub-sampled seismic data by giving up linearity and using an adaptive non-linear model. Convolutional neural networks (CNNs) are a novel strategy for the reconstruction of missing traces in pr-stack seismic images [17]. A new framework for training artificial neural networks (ANNs) was presented by Mikhailiuk and Faul [18] to restore corrupted multidimensional seismic signals. Wang et al. [19] utilized an eight-layer residual learning network (ResNet) with better back-propagation for interpolation.

The key aspect of rank reduction-based methods is that linear events in a clean seismic dataset are low-rank in the time and frequency domains. On the other hand, noise and missing traces increase the rank of data [20]. In the Fourier domain, rank reduction algorithms reduce the rank of Hankel/Toeplitz matrices generated from frequency slices. The singular spectrum analysis (SSA) method proposed by Sacchi [21], Tricket and Burroughs [22], Oropeza and Sacchi [23], and Bayati and Siahkoohi [24] works by rank-reduction of the Hankel matrix with an iterative algorithm in the frequency domain. Gao et al. [25] extended the SSA method to higher seismic data dimensions and called it a multichannel singular spectrum (MSSA). Interpolating regularly missing traces with a de-aliased MSSA method was proposed by Naghizadeh and Sacchi [26]. Kreimer et al. [27] developed the algorithm for pre-stack data via low-rank tensor completion. Kreimer and Sacchi [28] used the higher-order SVD for rank-reduction of the pre-stack seismic data tensor. Ely et al. [29] utilized a statistical test to control the complexity of regularization by tuning a single regularization parameter. Kumar et al. [30] developed rank-reduction techniques using matrix completion (MC) for seismic data reconstruction. Rekapali et al. [31] studied the application of multi-channel time slice singular spectrum analysis (MTSSSA) for 3D seismic data de-noising in the time domain and discussed the selection of the input processing window length. A damped OptShrink was studied by Siahshahr et al. [32], which selects the rank of the block of the Hankel matrix without knowing the final rank. Carozzi and Sacchi [33] proposed a method called (I-MSSA) that interpolates seismic data in its exact data coordinates that overcomes the problem of vertical errors in rough binning.

One of the necessary assumptions of rank-reduction-based techniques is that the Hankel matrix generated from clean and complete seismic data containing plane waves is low-rank, and its rank is equal to the number of plane waves present in the data. However, gaps and noise increase the rank, breaking this assumption. When dealing with data from complex geologies, rank selection becomes difficult and inconvenient. One approach to deal with this problem is to apply the algorithm to local windows. However, in most cases, it is not easy to find the proper window size because it is hard to decide if the structure in the local window is linear or not. Moreover, it is difficult to approximate the rank of each window. Choosing the wrong rank will lead to failure because, if overestimated, there will be a significant amount of residual noise remaining, and if underestimated, it will cause random noise and signal distortion.

In the presence of noise when the signal-to-noise ratio is low, data reconstructed by TSVD tend to contain a significant amount of residual noise. This residual noise is because part of the noise has a projection on the signal component projection. Nadakuditi [34] introduced an algorithm that provides a weighted approximation, where the weights reduce and threshold the singular values. Their method mitigates the effect of rank overestimation. If the rank is properly estimated, the algorithm will better estimate weak components of the signal subspace. Chen et al. [35] introduced a damping operator that shrinks the singular values containing significant particles of residual noise. Their method reached better results than the Cadzow rank-reduction method [22,23] in the reconstruction of highly noisy incomplete 5-D data. In this paper, we propose a method that automatically selects a rank for each local window. The method was proposed by Wu and Bai [36] for 2-D data. We improved it for 3-D data and propose an adaptive weighting rank reduction method (AWRR) that selects the rank of the block of Hankel matrices automatically to reconstruct and denoise 3-D seismic data simultaneously. The strategy is to choose the second cutoff in the singular-value spectrum of the block Hankel matrix. We tested the efficiency of the proposed method by applying it to both synthetic and real seismic data.

## 2. Materials and Methods

Consider $S\left(t,x,y\right)$ a block of 3-D seismic data in the t−x−y domain on $N_{t}$ by $N_{x}$ by $N_{y}$ samples. $\left(t=1,\cdots ,N_{t}\right)$, $\left(x=1,\cdots ,N_{x}\right)$, $\left(y=1,\cdots ,N_{y}\right)$. In the frequency domain, the data are represented as $S\left(\omega ,x,y\right)$ and $\left(\omega =1,\cdots ,N_{\omega }\right)$. Each frequency slice of the data at a given frequency $\omega_{m}$ can be represented by the following matrix:(1)$$S\left(\omega_{m}\right)=\left(\begin{array}{cccc} S\left(1,1\right) & S\left(1,2\right) & \cdots & S\left(1,N_{x}\right)\\ S\left(2,1\right) & S\left(2,2\right) & \cdots & S\left(2,N_{x}\right)\\ \vdots & \vdots & \ddots & \vdots \\ S\left(N_{y},1\right) & S\left(N_{y},2\right) & \cdots & S\left(N_{y},N_{x}\right) \end{array}\right).$$

To avoid notational confusion, let us ignore the argument $\omega_{m}$. Then, construct a Hankel matrix from each inline of S; for the inline *i*th, the Hankel matrix in the *m*th frequency slice will be:(2)$$H_{i}=\left(\begin{array}{cccc} S\left(i,1\right) & S\left(i,2\right) & \cdots & S\left(i,l\right)\\ S\left(i,2\right) & S\left(i,3\right) & \cdots & S\left(i,l+1\right)\\ \vdots & \vdots & \ddots & \vdots \\ S\left(i,N_{x}-l+1\right) & S\left(i,N_{x}-l+2\right) & \cdots & S\left(i,N_{x}\right) \end{array}\right).$$

At this step, we have $N_{y}$ Hankel matrices of each inline. Then, the multichannel singular spectrum analysis (MSSA) constructs a block Hankel matrix M from Hankel matrices $H_{i}$:(3)$$M=\left(\begin{array}{cccc} H_{1} & H_{2} & \cdots & H_{n}\\ H_{2} & H_{3} & \cdots & H_{n+1}\\ \vdots & \vdots & \ddots & \vdots \\ H_{N_{y}-n+1} & H_{N_{y}-n+2} & \cdots & H_{N_{y}} \end{array}\right).$$

The size of M is I×J, where $I=\left(N_{x}-m+1\right)\left(N_{y}-n+1\right)$ and J=mn. The integers *m* and *n* are chosen to make the Hankel matrices of H and the block Hankel matrix of M square matrices, or close to square. As we see, including the two spatial dimensions of the 3-D cube for each frequency can make the block Hankel matrix very large.

In seismic data, the observed data can be indicated as $S_{obs}=R\left(S_{0}+\eta \right)$, where $S_{obs}$ represents observed seismic data, $S_{0}$ indicates the full noiseless data, η represents the random noise and the residuals, and R indicates the sampling matrix formed of zeros and ones. Using the MSSA algorithm, we can write the Hankel matrix of observed data as:(4)M=P+N,
where P represents low-rank Hankel matrix of the desired signal and N denotes the noise component which includes the gaps as well. In MSSA, the decomposition of the Hankel matrix in the rank-reduction step using TSVD will be as follows:(5)$$M=U\Sigma V^{H}.$$

If one knows the desired rank of the Hankel matrix, the estimated signal desired signal is recovered by:(6)$$\hat{M}=U_{k}\Sigma_{k}V_{k}^{H},$$
where $\hat{M}$ is the estimated signal, and *k* is the predefined rank of the Hankel matrix equal to the number of the linear events in each local window.

Let us investigate the effect of linear events on the singular values in each frequency. Figure 1a shows the cube of 3-D data. Figure 1b indicates an inline of the data and Figure 1c represents a slice of data in a cross-line direction.

We investigate the singular value distribution of the data by converting it to the $\left(f,x,y\right)$ domain. Next comes generating the block Hankel matrix in each frequency slice. Figure 2a shows the block Hankel matrix for the frequency slice of 20 Hz. Figure 2b displays the block Hankel matrix generated in the frequency slice of 50 Hz. According to Figure 2a,b, it is clear that the block Hankel matrix for the frequency of 20 Hz is smoother than the one for the frequency of 50 Hz. It signifies that the block Hankel matrices for the higher frequencies require a higher rank than those in the lower frequencies to recover.

The singular-value spectrum of the data’s block Hankel matrices for its frequency range is presented in Figure 3. Figure 3 represents the first 15 singular values’ spectrum. This figure illustrates how the number of nonzero singular values of the block Hankel matrix at low frequencies is equal to the number of linear events but increases with frequency.

Similarly, Figure 4a represents the bar plot of the normalized singular values for the frequency slice of 20 Hz. Figure 4b shows the zoomed image for the first 20 singular values. Figure 4c shows the bar plot of the singular values for the frequency slice of 50 Hz. Figure 4d is the zoomed image for the first 20 singular values. From the figures, one can see that for the frequency of 20 Hz, the maximum difference in the energy between two adjacent singular values occurs in the third singular value, and it is the same as the number of linear events. However, for the frequency slice of 50 Hz, there is a second group of nonzero singular values. It means that we need to keep more singular values to recover all the frequencies completely.

To analyze the effect of each singular value on the energy of each frequency for the data presented in Figure 1, data were decomposed into their singular matrices and recovered with each singular value from 1 to 15 separately. Then, the energy per frequency or power spectrum of the recovered data with each singular value is calculated. In Figure 5, the red line represents the graph of frequency energy per frequency, and the blue line shows the estimate of the mean normalized frequency of the power spectrum. Figure 5a is the result of recovering data with the first singular value. Figure 5b shows the energy per frequency for the recovered data with the second singular value. Figure 5c is the energy per frequency for the recovered data, including just the third singular value. The estimated mean frequency for these first three singular values is 77 Hz. Figure 5d–f represent the energy per frequency for the recovered data, including just the 4th, 5th, and 6th singular values, respectively. The estimated mean frequency for these singular values is 90 Hz. Figure 5g–i indicates the energy per frequency for the recovered data, including just the 7th, 8th and 9th singular values, respectively. The estimated mean frequency for these singular values is 88 Hz. Figure 5j–l show the energy per frequency for the recovered data, including just the 10th, 11th, and 12th singular values, respectively. It is clear that most energy of the useful signal is recovered with the first three singular values. Nevertheless, the shift in the mean frequency from 77 to 90 Hz states that there is leakage for the higher frequencies in the data. We conclude from this test that despite the remaining residual errors in the presence of the additive noise, we need to choose the rank of the block Hankel matrix to be greater than the number of linear events in each processing window of frequency and space.

To find the best rank of data that minimizes residual errors, we tested different kinds of data containing various numbers of events with different slopes and amplitudes. The best result is when we choose the rank from the second cutoff of singular values instead of the first cutoff.

In all tested data, it is possible to see the first cutoff with a distinct change in the energy of singular values. However, sometimes there is no abrupt change in the amplitude of the singular values for the second cutoff, especially in the presence of a high level of random noise. Finding the second cutoff could be a challenge when the quality of the signal is poor. With a careful look at where the first and the second cutoff in clean and complete data happen, we conclude that there is a linear relationship between the first and second cutoff. The best estimate for the second cutoff can be estimated as follows:(7)$$T\left(\Sigma ,k\right)=max_{i}\frac{\sigma_{i}^{2}}{\sigma_{i+1}^{2}},$$
(8)$$B\left(\Sigma ,\tilde{k}\right)=3\times T\left(\Sigma ,k\right),$$
where $\sigma_{i}$ is the *i*th singular value of the Hankel matrix in each frequency. $T\left(\Sigma ,k\right)$ indicates the operator that finds the rank at the point where the two following singular values become more scattered. $B\left(\Sigma ,\tilde{k}\right)$ indicates the second cutoff of the singular-value spectrum of the block of a Hankel matrix in each frequency slice. $\tilde{k}$ can be introduced as the optimal rank of the block Hankel matrix that minimizes the Frobenius-norm difference between the approximated and the exact signal components.

Substituting Equation (Equation 8) to the rank-reduction step will give us:(9)$$\tilde{M}=U_{\tilde{k}}B\left(\Sigma ,\tilde{k}\right)V_{\tilde{k}}^{H},$$

By applying Equation (Equation 9) on the rank-reduction step of MSSA, the rank-reduced block Hankel matrix is reconstructed. The next step is averaging anti-diagonals of the recovered block Hankel matrix. It recovers the signal in the Fourier domain for each frequency slice. The rank-reduction step leads to gradually reconstructing the missing traces and can be used together with an iterative algorithm to replace the missing traces with the reconstructed traces. An iterative algorithm is a practical approach for random noise attenuation and seismic data amplitude reconstruction. The algorithm is written as follows:(10)$$\begin{array}{c} {\tilde{S}}_{n+1}\left(\omega_{m}\right)=\alpha_{n}S_{obs}+\left(I-\alpha_{n}R\right)F^{-1}\mathrm{ARBF}\left({\tilde{S}}_{n}\left(\omega_{m}\right)\right),\;\;\;\;\;\;n=1,2,\cdots ,N \end{array}$$
where $\omega_{m}$ is the temporal frequency, I is a matrix of ones, $\alpha_{n}\in \left(0,1\right]$ is a weight factor, which decreases linearly with iterations, F indicates the Fourier transform, and $F^{-1}$ shows the inverse Fourier transform. B points to the Hankelization operator to generate the block Hankel matrix, R reveals the rank reduction operator, and A gives the averaging anti-diagonal operator.

Selecting the rank using Equation (Equation 9) recovers all the signal components. One of the advantages of this rank-reduction method is that it is adaptive and data-driven, and there is no need to set any parameter for the rank-reduction step in each processing window. Moreover, choosing the second cutoff instead of the first one leads to a better reconstruction of the high frequencies. However, selecting large ranks leads to more residual errors in the recovered data. That is why we need to apply a weighting operator to reduce the effect of noise on the projected signal components to recover higher frequencies.

We are looking for a weighting operator $\hat{W}$ to adjust the singular values of $\tilde{M}$ to calculate the best estimation of the desired signal. Nadakuditi [34] proposed an algorithm for low-rank matrix denoising that can be summarized as follows
$M_{n\times m}$ is the signal plus noise Hankel matrix.*k* is the best effective rank that can represent the signal.For i=1:k,
(11)$$\mathrm{compute}\;{\hat{w}}_{i}=\left(-\frac{2}{\sigma_{i}}\frac{D\left(\sigma_{i};\Sigma \right)}{D^{\prime }\left(\sigma_{i};\Sigma \right)}\right),$$End for loop,Compute results as $\hat{M}=\sum_{i=1}^{k}{\hat{w}}_{i}\sigma_{i}u_{i}v_{i}^{H}$

In this algorithm, $D\left(\sigma ;\Sigma \right)$ is computed as:(12)$$\begin{array}{ccc} D\left(\sigma ;\Sigma \right) & =\frac{1}{k}Tr\left(\sigma {\left(\sigma^{2}I-\Sigma \Sigma^{H}\right)}^{-1}\right)\frac{1}{k}Tr\left(\sigma {\left(\sigma^{2}I-\Sigma^{H}\Sigma \right)}^{-1}\right) & ={\left[\frac{1}{k}Tr\left(\sigma {\left(\sigma^{2}I-\Sigma^{2}\right)}^{-1}\right)\right]}^{2}, \end{array}$$
where D represents D-transform and $Tr\left(.\right)$ denotes the trace operator of the input.

The $D^{\prime }$ represents the derivative of D with respect to σ:(13)$$\begin{array}{cc} D^{\prime }\left(\sigma ;\Sigma \right) & =2\left[\frac{1}{k}Tr\left(\sigma {\left(\sigma^{2}I-\Sigma^{2}\right)}^{-1}\right)\right]\\ & =\left[\frac{1}{k}Tr\left({\left(\sigma^{2}I-\Sigma^{2}\right)}^{-1}-2\sigma {\left(\sigma^{2}I-\Sigma^{2}\right)}^{-2}\sigma \right)\right]\\ & =\frac{2}{k^{2}}\left[Tr\left(\sigma {\left(\sigma^{2}I-\Sigma^{2}\right)}^{-1}\right)\right]\left[Tr{\left({\left(\sigma^{2}I-\Sigma^{2}\right)}^{-1}-2\sigma^{2}{\left(\sigma^{2}I-\Sigma \right)}^{2}\right)}^{-2}\right)], \end{array}$$

The *D*-transform describes how the distribution of the singular values of the sum of the independent matrices is related to the distribution of the singular values of the individual matrices [37]. Benaych and Nadakuditi [38] indicate that the principal singular values and vectors of a large matrix can be set apart as the singular values of the signal matrix and the D-transform of the limiting noise-only singular value distribution. From Equation (Equation 11), the weighting operator $\hat{W}$ can be written as:(14)$$\hat{W}=diag\left({\hat{w}}_{1},{\hat{w}}_{2},\cdots ,{\hat{w}}_{k}\right).$$

We can substitute the weighting algorithm obtained from Equation (Equation 14) into Equation (Equation 9) to enhance the results of the rank-reduction step as:(15)$$\hat{M}=U_{\tilde{k}}\hat{W}B\left(\Sigma ,\tilde{k}\right)V_{\tilde{k}}^{H},$$
where $\hat{M}$ indicates the reduced rank block Hankel matrix. Missing traces can be interpolated completely by applying the iterative algorithm. This adaptive-weighting rank-reduction (AWRR) method leads the way that sorts out the rank of the block Hankel matrix automatically while reducing the effect of the residual errors.

The weighting operator satisfies the equation below [34]:(16)$$∥\hat{M}-P∥\leq \epsilon .$$

To understand the effect of the weighting operator on the singular values of the block Hankel matrices, the predefined rank-reduction method (TRR) and the weighting rank-reduction method (WRR) with predefined rank were applied to a block Hankel matrix. Figure 6a shows clean and complete 3-D synthetic data having four linear events. Figure 6b indicates the same 3-D data with SNR=2 and 51% missing traces. Figure 6c,d show slices of data in inline directions for the cubes of Figure 6a and Figure 6b, respectively. Figure 6e,f are the slices of the cubes of Figure 6a and Figure 6b in the crossline direction, respectively.

Figure 7 shows the Hankel matrices of data in Figure 6 for the frequency slice of 20 Hz. Figure 7a is the block Hankel matrix of clean and complete data. Figure 7b corresponds to the block Hankel matrix of data with SNR=2 and 51% missing traces. Figure 7c shows the block Hankel matrix after applying the predefined rank = 12 for ten iterations. Figure 7d indicates the block Hankel matrix after applying the weighting operator to adjust the singular values of the data using TSVD with predefined rank = 12 after ten iterations. We can see from Figure 7 that the block Hankel matrix of the WRR method is smoother than the result of applying a predefined rank = 12 after ten iterations.

Figure 8 corresponds to the spectrum of the first 25 singular values of the block Hankel matrices of Figure 6. Figure 8a represents the singular-value spectrum of the block Hankel matrix of clean and complete data for the frequency of 20 Hz, the first three singular values indicate the useful signal that relates to the coherent events in the data. Figure 8b relates to the singular-value spectrum of the noisy and incomplete data, Figure 8c shows the singular-value spectrum of the noisy and incomplete data after applying TSVD with rank = 12. Figure 8d indicates the singular-value spectrum of the noisy and incomplete data after applying TSVD with rank = 12 and the weighting operator. We can see that the singular spectrum of the data after applying the weighting operator is much closer to the spectrum of the desired signal in Figure 8a.

Figure 9 corresponds to the block Hankel matrices of the same data in the constant frequency slice of 50 Hz. Figure 9a shows the block Hankel matrix of the clean and complete data, and Figure 9b is the block Hankel matrix of data with SNR=2 where 51% of traces are removed. Figure 9c displays the recovered block Hankel matrix after applying TRR and rank = 4 for ten iterations. Figure 9d indicates the block Hankel matrix after applying the WRR with predefined rank = 4. Figure 9e represents the recovered block Hankel matrix of the ARR method, and Figure 9f is the block Hankel matrix recovered by the weighting ARR (AWRR) method. Generally, the results of AWRR and WRR are smoother than the output of TRR and ARR, and the results of ARR and AWRR include more details than those of TRR and WRR.

Figure 10 corresponds to the spectrum of the first 25 singular values of the block Hankel matrices of Figure 9. Figure 10a represents the singular-value spectrum of the block Hankel matrix of clean and complete data for the frequency of 50 Hz. The abrupt drop in energy of the third and the fourth singular values relates to the useful signal. However, there are still nonzero singular values that are related to the useful signal. Figure 10b is the singular-value spectrum of the block Hankel matrix of noisy and incomplete data. Figure 10c shows the singular-value spectrum of the noisy and incomplete data after TRR with rank = 4. Figure 10d displays the singular-value spectrum of the noisy and incomplete data after applying WRR with rank = 4. Figure 10e depicts the singular-value spectrum after applying ARR. Figure 10d indicates the singular-value spectrum of the noisy and incomplete data after the implementation of the AWRR method. From the figures, one can see that the result of AWRR is more comparable to the corresponding result for the desired signal in Figure 10a.

## 3. Results and Discussion

Several evaluations are presented in this section to estimate the proficiency of the different methods of rank reduction for the MSSA algorithm. First, the methods were tested on a synthetic shot gather containing nine hyperbolic events with different curvatures. Then, they were tested on a shot gathered from a 3-D field dataset. To judge the reconstruction results numerically, we use an interpolation quality factor (QF) defined by:(17)$$QF=10\log_{10}\left(\frac{∥d_{0}{∥}_{2}^{2}}{∥d_{f}-d_{0}{∥}_{2}^{2}}\right),$$
where $d_{0}$ is the clean and complete data, and $d_{f}$ is the result after applying an interpolation algorithm.

### 3.1. Synthetic Data

Figure 11 shows the result of applying the previously discussed methods of rank-reduction to a synthetic shot gather holding nine hyperbolic events with different curvatures. The first test is a cube of 100 inline and 11 crosslines with SNR=2 and 60% decimated traces. We chose the local window with 23 traces in the inline direction and 11 in the crossline direction for each method and half a window overlapping in each direction. We set the number of iterations to be constant for all of the methods and the frequency range for the application of the MSSA algorithm to 1 to 100 Hz. The rank of TRR and WRR was set to nine. Figure 11a is the desired data arranged into a 2-D matrix. Figure 11b shows input data with SNR=2 and 60% missing traces. The results of applying TRR, ARR, WRR, and AWRR methods are shown in Figure 11c,e,g,i, respectively. Figure 11d,f,h,j are residual errors of Figure 11c,e,g,i, respectively. All four methods recovered the signals with correct amplitudes. However, in the TRR and ARR results, we can see significant residual errors. The AWRR result is much cleaner than those of TRR, ARR, and WRR. The input data quality factor is *QF* = −1.44 dB. The output quality factors of the TRR, ARR, WRR, and AWRR methods were *QF* = 5.23, 6.58, 7.56, 8.12 dBs, respectively.

Figure 12 compares the f−k spectra of the discussed rank-reduction methods. Figure 12a shows the f−k spectra of clean data. Figure 12b represents the f−k spectra of data with SNR=2 and 50% missing traces. Figure 12c–f are the f−k spectra of the TRR, WRR, ARR, and AWRR methods, respectively.

To evaluate the stability of each algorithm, we ran them with different percentages of gaps and noise realizations for an SNR of two. The other parameters were constant for each run. Figure 13a shows a graph of the interpolation quality factor for each method versus the gap ratio. The length of the error bars is the standard deviation of the interpolation quality factor, and each colored line connects the mean value of the interpolation quality factor for each method at each gap ratio. As we can see in Figure 13a, the input quality factor decreases with increasing gap ratio, and the output quality factor decreases with increasing gap ratios. Moreover, the output quality factor curve of the AWRR method is always above the other curves.

In the next experiment, the stability analysis was repeated to test the sensitivity of each method to the additive noise. The test was performed by changing the level of signal-to-noise ratio with 20 different realizations of SNR for a gap ratio =50%. We set the other parameters to be constant for each run. In Figure 13b we can see that the proposed method (AWRR) outperforms the other methods even with poor quality of the signal.

Table 1 indicates the values of the mean and standard deviation of the interpolation quality of each method for each gap percent. We can see that the average value of the AWRR quality factor is higher than for the other methods. Regarding the discussed methods, AWRR has lower standard deviation values, which indicates the values tend to be close to the mean value, especially when the data are sparser.

Table 2 represents the values of the mean and standard deviation of the output quality factor of each method for each signal-to-noise ratio input.

### 3.2. Real Field Data

This experiment tested the efficiency of the methods on a shot gathered of a 3-D field dataset. It is easy to inspect traces in shot/receiver gather displays for poor receivers or any bad shots, which are the logistic constraints during the seismic survey. Regarding the input data, the best results are obtained with NMO-corrected data [6]. However, in this experiment, we applied the MSSA interpolation before NMO correction to see the effect of the algorithm on the curvature.

Figure 14 refers to the acquisition coordinates of the shot gather. The red star represents the location of the shot, and the black dot indicates the receiver’s location. Figure 14a shows the initial distribution of the traces in a shot gather. For this experiment, 41% of the traces were removed. Figure 14b illustrates the geometry of input traces after removed 41% of them.

This test applied the MSSA algorithm using the AWRR and TRR methods in the rank-reduction step. Figure 15a shows the input cube of data, which presents the missing traces. Figure 15b shows the result of the interpolation using TRR in the rank-reduction step, and Figure 15c shows the result of using AWRR. For both tests, the number of iterations and processing window remained unchanged. For the TRR approach, the predefined rank wa set to rank = 10. Figure 16 represents the cube of data arranged into a 2-D matrix. Figure 16a shows the input data. Figure 15b shows the result of applying TRR. Figure 16c shows the result of applying AWRR. Figure 17 corresponds to a patch of data from the time 1.35 (s) to 1.75 (s) and the trace numbers 160 to 190. Figure 17 represents the cube of data rearranged in a 2-D matrix. Figure 17a shows the input data. Figure 17b shows the result of applying TRR. Figure 17c shows the result of applying AWRR.

## 4. Conclusions

We proposed an adaptive weighting rank-reduction (AWRR) method by exploring the singular values of the Hankel matrix in each frequency slice. The method chooses the maximum energy ratio between the two following singular values as the criterion to define the optimal rank. For 3-D data, the block Hankel matrices in higher frequencies are recovered with higher ranks. Consequently, the method selects the second cutoff in the singular-value spectrum. AWRR is based on an adaptive rank-reduction method combined with a cascade weighting operator. It is adaptive in selecting the rank of the block Hankel matrices. In addition, the weighting operator reduces the effect of additive random-noise projection on data, so a smoother block of Hankel matrices is its result. Furthermore, for different types of data, such as linear and curved, through a lot of simulations with different SNR and gap percentages, the proposed algorithm showed a higher output SNR and a better reconstruction of data with the predefined rank reduction methods. Most interpolation rank-reduction methods, such as MSSA, are SVD-based methods in their rank-reduction step. An SVD decomposition of a Hankel matrix is a very time-consuming operation. In addition, the iterative approach in the interpolation part of MSSA makes the algorithm time-consuming. To make AWRR in the MSSA algorithm applicable for higher dimensions, it is recommended to move from the SVD-based method to the SVD-free-based rank minimization method.

## Figures and Tables

**Figure 1 sensors-23-00577-f001:**
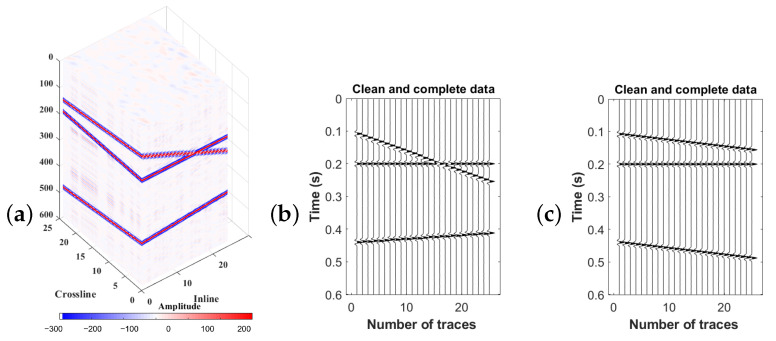
(**a**) Cube of 3-D data having three linear events. Slice of data in (**b**) inline direction, (**c**) cross-line direction.

**Figure 2 sensors-23-00577-f002:**
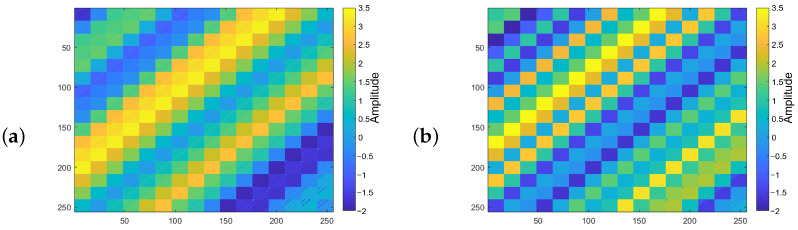
Block Hankel matrix for frequency slice of (**a**) 20 Hz, (**b**) 50 Hz. The color bar represents the absolute value of the FFT of input data in the mentioned frequency.

**Figure 3 sensors-23-00577-f003:**
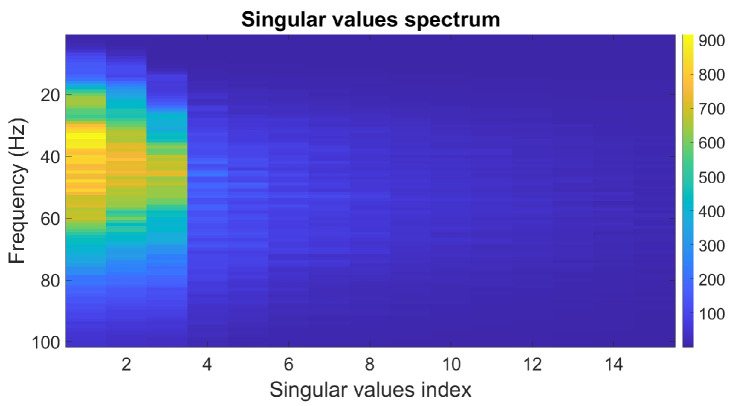
Singular-value spectrum. The color bar represents the magnitude of the singular values.

**Figure 4 sensors-23-00577-f004:**
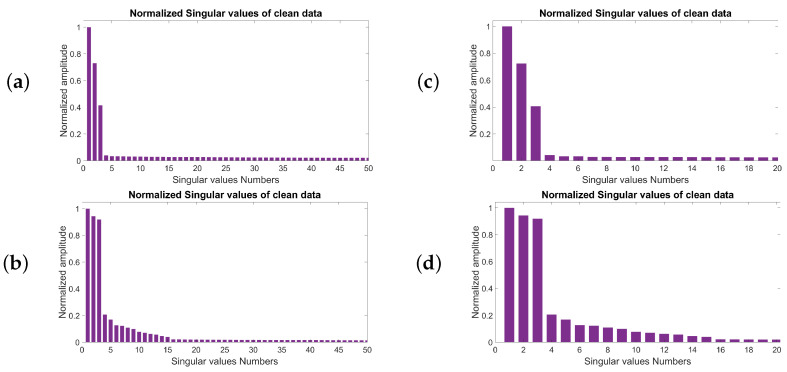
Normalized singular value distribution of the block Hankel matrix of 3-D data with three linear events in the frequency slice: (**a**) 20 Hz, (**b**) 50 Hz. The first 20 singular values of the same data for frequency slice of (**c**) 20 Hz and (**d**) 50 Hz.

**Figure 5 sensors-23-00577-f005:**
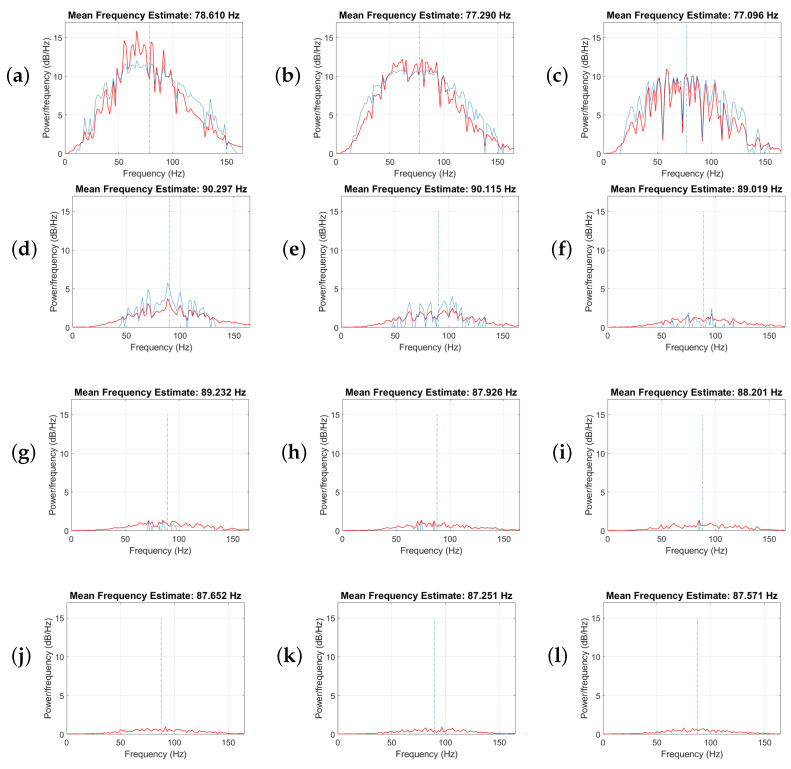
Analysis of the power spectrum for each singular value. Panels (**a**–**l**) show the power spectrum recovered by the 1st to 12th singular values, respectively.

**Figure 6 sensors-23-00577-f006:**
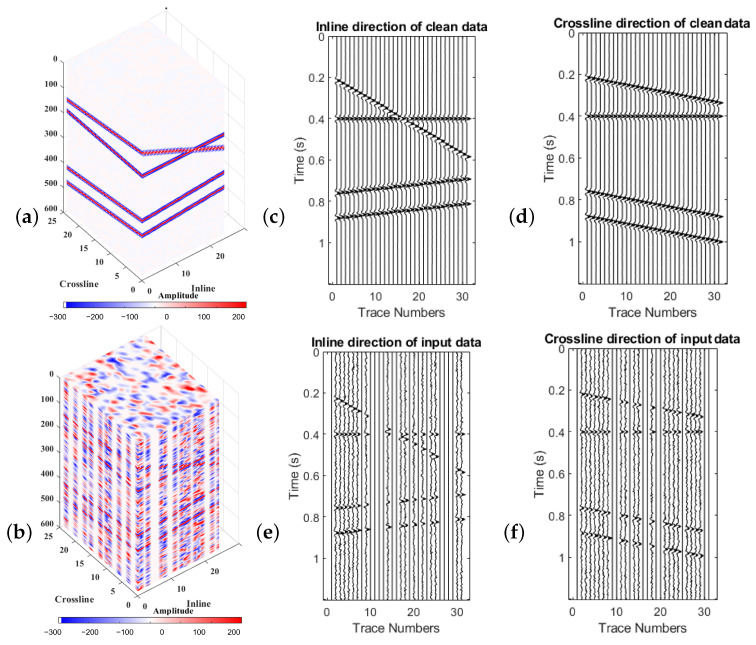
(**a**) Cube of 3-D synthetic clean and complete data; (**b**) Cube of 3-D synthetic data with SNR=2 and 51% missing traces; (**c**,**d**) inline and cross-line sections of cube (**a**), respectively; (**e**,**f**) inline and cross-line sections of cube (**b**), respectively.

**Figure 7 sensors-23-00577-f007:**
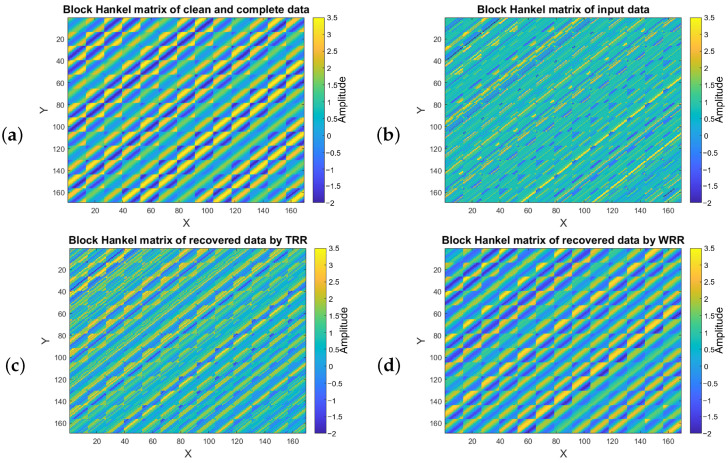
Block Hankel matrices for frequency of 20 Hz for (**a**) clean and complete data, (**b**) data with SNR=2 and 51% missing traces, (**c**) data recovered by TSVD rank-reduction after ten iterations, (**d**) data recovered by WRR after ten iterations.

**Figure 8 sensors-23-00577-f008:**
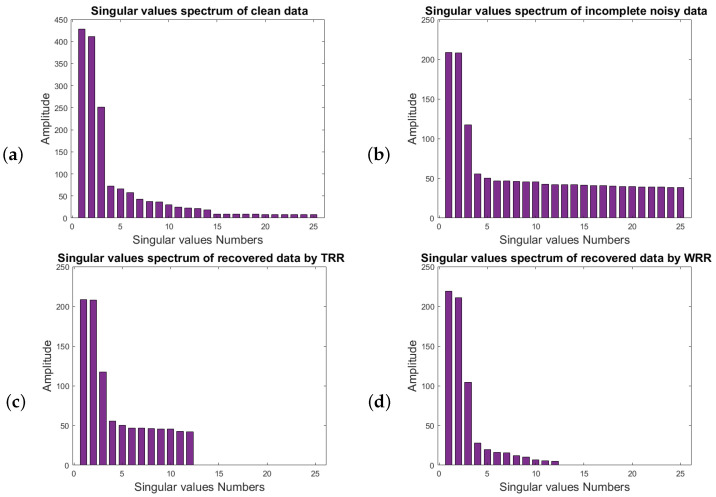
The spectrum of the first 25 singular values of the block Hankel matrices for frequency of 20 Hz for (**a**) clean and complete data and (**b**) data with SNR=2 and 51% missing traces; (**c**) the first 12 singular values of the data; (**d**) singular-value spectrum of the data after applying the weighting operator.

**Figure 9 sensors-23-00577-f009:**
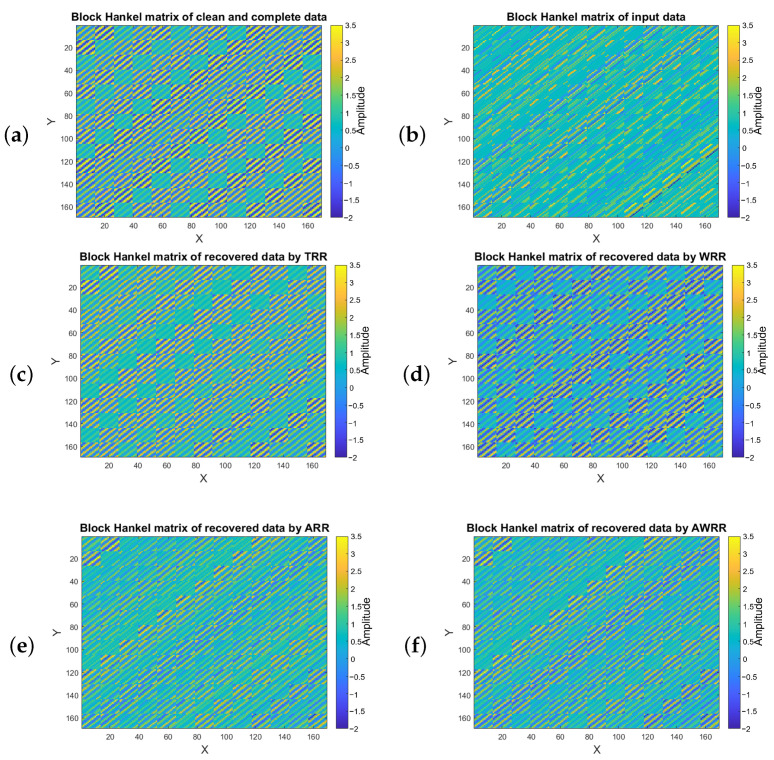
Block Hankel matrices for frequency of 50 Hz for (**a**) clean and complete data and (**b**) data with SNR=2 and 50% missing traces; (**c**) block Hankel matrix recovered by TRR rank = 4 after ten iterations; (**d**) data recovered by WRR after ten iterations. (**e**) Data recovered by applying the ARR method for ten iterations, and (**f**) data recovered after applying AWRR for ten iterations.

**Figure 10 sensors-23-00577-f010:**
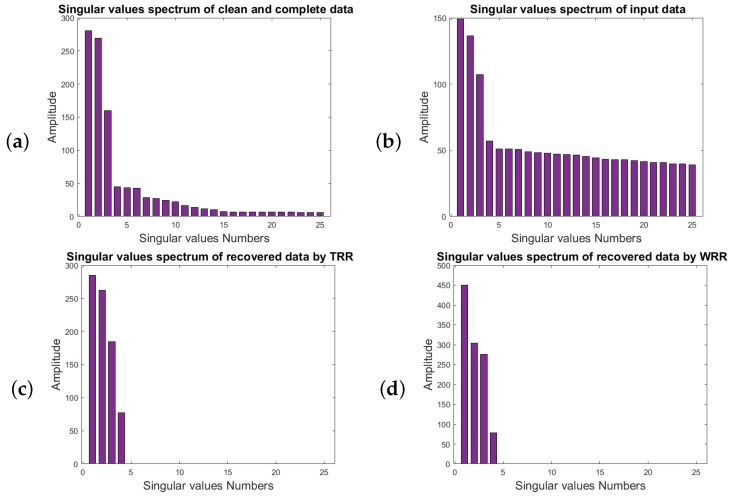
The spectrum of the first 25 singular values of the block Hankel matrices for frequency slice of 50 Hz for (**a**) clean and complete data, (**b**) data with SNR=2 and 50% killed traces, (**c**) data recovered by TRR, (**d**) data recovered by WRR, (**e**) data recovered by ARR, (**f**) data recovered by AWRR.

**Figure 11 sensors-23-00577-f011:**
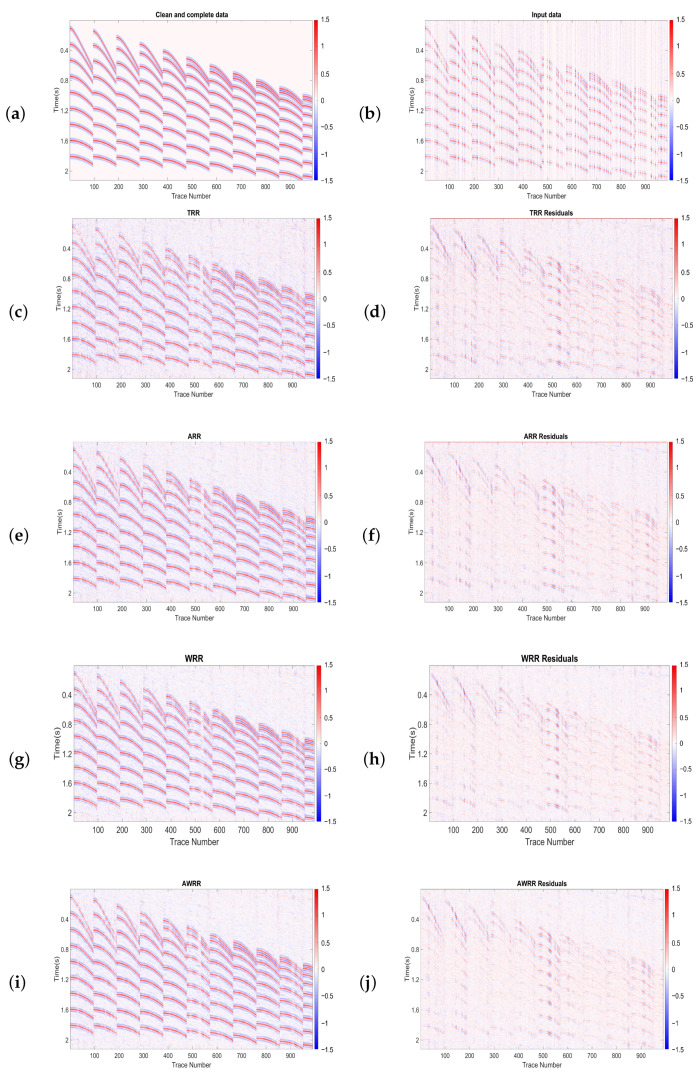
Comparison of different methods of rank-reduction. (**a**) Clean and complete data, (**b**) input data with SNR=2 and 60% missing traces; (**c**,**e**,**g**,**i**) data interpolated by TRR, ARR, WRR, and AWRR, respectively; (**d**,**f**,**h**,**j**) residual errors of TRR, ARR, WRR, and AWRR, respectively.

**Figure 12 sensors-23-00577-f012:**
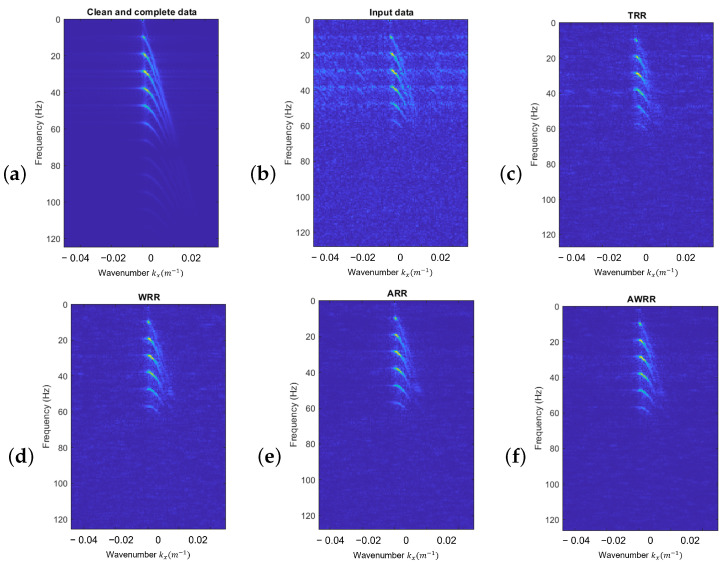
f−k spectral comparison of (**a**) clean and complete data, (**b**) input data, (**c**) TRR result, (**d**) WRR result, (**e**) ARR result, (**f**) AWRR result.

**Figure 13 sensors-23-00577-f013:**
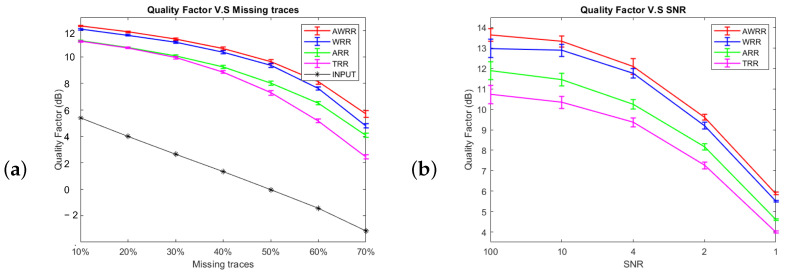
(**a**) The mean and standard error of the quality of reconstruction versus gap ratio for various methods. These results were obtained by running each method on 20 noise realizations of the dataset with different gap ratios and rank = 9 for the methods requiring a predefined rank. (**b**) The mean and standard error of the reconstruction quality versus SNR for various methods. These results were obtained by running each algorithm on 20 noise realizations of the dataset with a gap ratio =50%, and rank = 9 for the methods requiring a predefined rank.

**Figure 14 sensors-23-00577-f014:**
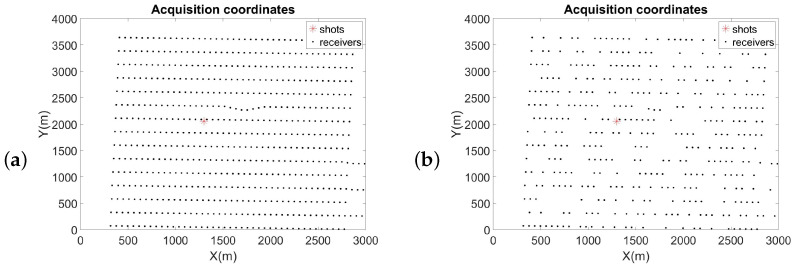
Geometry of the field data. (**a**) Initial distribution of the traces. (**b**) Distribution of traces after removing 41% of them.

**Figure 15 sensors-23-00577-f015:**
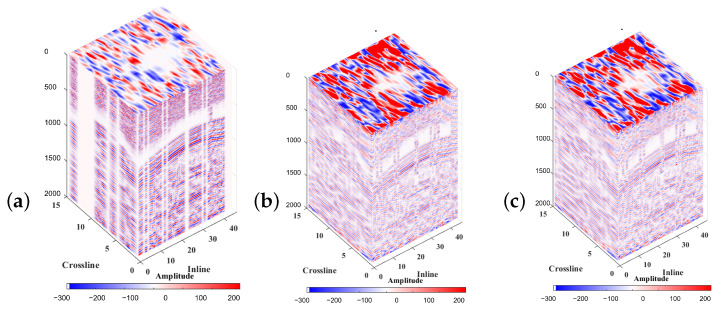
Cube of 3-D field data. (**a**) The input data. (**b**) Interpolation result of applying TRR. (**c**) Interpolation result of applying AWRR.

**Figure 16 sensors-23-00577-f016:**
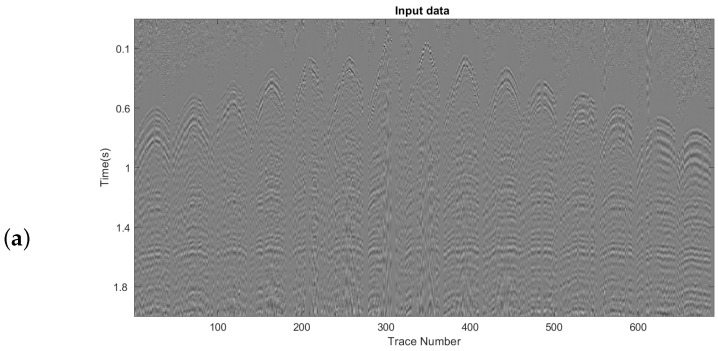
(**a**) Input real data, (**b**) interpolation result of applying TRR approach, (**c**) interpolation result of applying the AWRR method.

**Figure 17 sensors-23-00577-f017:**
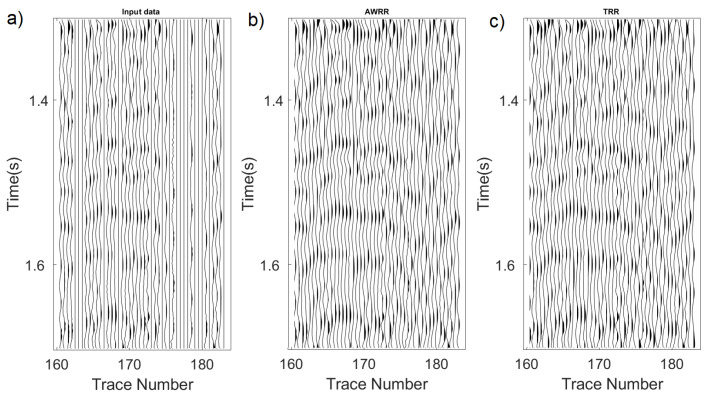
Zooming area of Figure 16 from time 1.35 (s) to 1.75 (s) and trace numbers 160 to 190. (**a**) input data, (**b**) TRR result, (**c**) AWRR result.

**Table 1 sensors-23-00577-t001:** Mean and standard deviation of the quality of interpolation for each method for different gap-ratio inputs.

	Output Quality Factor (dB)							
	AWRR		WRR		ARR		TRR	
**Gap Ratio**	**MEAN (dB)**	**STD**	**MEAN (dB)**	**STD**	**MEAN (dB)**	**STD**	**MEAN (dB)**	**STD**
10%	12.34	0.04	12.10	0.04	11.22	0.06	11.17	0.04
20%	11.90	0.05	11.64	0.05	10.71	0.09	10.69	0.06
30%	11.36	0.05	11.12	0.05	10.08	0.08	8.86	0.08
40%	10.45	0.07	10.35	0.07	9.24	0.10	8.90	0.11
50%	9.64	0.16	9.38	0.20	8.01	0.17	7.28	0.20
60%	8.13	0.13	7.61	0.13	6.51	0.14	5.16	0.20
70%	5.70	0.16	4.78	0.21	4.10	0.21	2.46	0.25

**Table 2 sensors-23-00577-t002:** Mean and standard deviation of the quality of interpolation for each method for different SNR inputs.

	Output Quality Factor (dB)							
	AWRR		WRR		ARR		TRR	
**Input SNR**	**MEAN (dB)**	**STD**	**MEAN (dB)**	**STD**	**MEAN (dB)**	**STD**	**MEAN (dB)**	**STD**
100	13.64	0.41	12.98	0.41	11.90	0.44	10.72	0.30
10	13.32	0.42	12.89	0.42	11.44	0.30	10.34	0.28
4	12.10	0.26	11.77	0.28	10.25	0.23	9.36	0.38
2	9.61	0.26	9.20	0.26	8.16	0.16	7.26	0.15
1	5.88	0.09	5.51	0.09	4.61	0.05	3.40	0.07

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
