# Peer review of "3-D Data Interpolation and Denoising by an Adaptive Weighting Rank-Reduction Method Using Multichannel Singular Spectrum Analysis Algorithm"

_sensors, 2023, doi:10.3390/s23020577_

Round 1

Reviewer 1 Report

Reviewer’s report on manuscript entitled:

3-D data Interpolation and denoising by adaptive weighting rank-reduction method using multichannel singular spectrum analysis algorithm

The authors propose a rank-reduction method for seismic data interpolation and de-noising. The methodology and results are interesting and sound, however, the presentation needs improvement. Below, please see my comments. 

Abstract needs to be improved.

Please clearly state the objective of the manuscript. You proposed a method for 3D seismic data interpolation and denoising. This should be clearly mentioned in the Abstract.

Line 4. Please define SVD. Please note that all the abbreviations must be defined in both Abstract and Introduction.

Lines 29, 36, etc. please don’t start a sentence with reference number. For example in line 29, you can say “Stolt [11] introduced…” please check and modify elsewhere.

line 47. “increase the rank”? Please clarify.

Lines 50-55. There are other seismic data interpolation and denoising that can be mentioned here. For example, Multichannel Anti-Leakage Least-Squares Spectral Analysis (MALLSSA) and Multichannel Interpolation by Matching Pursuit (MIMAP)

https://doi.org/10.1007/s11600-019-00320-3

https://doi.org/10.1190/1.3496958

Line 85. Please use bullet points to state the main contributions of this article at the end of Introduction. 

Line 86. This is a section. Please see the MDPI guideline. The structure of the manuscript is recommended to be as follows:

1. Introduction 

2. Materials and Methods

2.1 Study region and dataset 

here please describe your simulated and field datasets and geographical location of region.

2.2 Weighted rank selection

You may also use subsections here to organize the method section better.

3. Results and discussion 

4.  Conclusions 

Figure 2. Please insert color bar with label and unit

Figure 3 Please insert the label and unit for the color bars

Figure 5. Caption. Please simply say:

Panels (b) to (l) show power spectrum recovered by the 2nd to 12th singular value, respectively.

Format issue. When you refer to equations, please use parentheses. For example, equation (9) not equation 9.

Figure 11. I am assuming that you used a fix data range for generating the colours in all the panels to visually compare the performance of each method. I suggest to add one color bar at the bottom of Figure 11 and insert its label. Also, it would be nice to show the wiggles plot of a zoomed (magnified) section of this figure in another figure, so the reader can see the comparisons more clearly. Furthermore, I think adding f-k spectra plots (frequency-wavenumber)  oils also be useful for comparing the results of each method and observing alias events.

Figures 16 and 17. Please show the corresponding f-k spectrum on the right hand side of each panel for visualization and comparison purposes. Please use the same value range for f-k spectra and same value range for the seismic data. 

The conclusion part is short and can be improved. Please discuss the computational complexity of the applied method and your proposed method. Please add the limitations of the study and method.

Please follow the MDPI guideline for formatting the references. For example, References 13-16 have different style for volume, page number, etc. 

Thank you for your contribution 

Author Response

Dear Reviewer,

We would like to say thank you for the valuable comments to improve the paper. We have managed all the comments as explained below.

First, we would like to say thank you to the reviewers for the useful comments to improve the paper. We have addressed all the comments in the attached pdf file.

Regards,

Farzaneh Bayati

Reviewer 2 Report

The authors propose the AWRR algorithm based on a weighting operator and adaptive rank-reduction method. The effectiveness of the method is verified by comparing the result of different methods. Overall, the paper is well organized and its presentation is also good. However, the following comments and suggestions should be considered:

1. This paper discusses the influence of frequency on the number of singular values. Higher frequency results in more singular values. However, the influence of the dominant frequency of the wavelet is not discussed. When the dominant frequency of the wavelet is 20 Hz, is the number of singular values of 20 Hz equal to 3 for three linear events?

2. How to protect the weak but effective signals with random noise using the AWRR?

3. In Fig. 6 (b), (c), (d), (e) on Page 8, the captions are inconsistent with the text.

4. In Fig. 11, the interpolation effect with a big gap needs to be improved.

5. For the block segmentation described in lines 143-144, it is suggested that authors should take into account the effect of block size on the interpolation.   

6. According to the examples of the synthetic and the real field data, the aliasing caused by the Fourier transform still exists.

 In addition, two relevant references should be considered in your introduction.

[1] Rajesh, R., Tiwari, R.K., Sen, Mrinal K., Vedanti, Nimisha, 3D seismic data de-noising and reconstruction using Multichannel Time Slice Singular Spectrum Analysis, Journal of Applied Geophysics (2017), doi:10.1016/j.jappgeo.2017.04.001

[2] M. A. Nazari Siahsar, S. Gholtashi, E. Olyaei Torshizi, W. Chen and Y. Chen, Simultaneous Denoising and Interpolation of 3-D Seismic Data via Damped Data-Driven Optimal Singular Value Shrinkage, IEEE Geoscience and Remote Sensing Letters, vol. 14, no. 7, pp. 1086-1090, July 2017, doi: 10.1109/LGRS.2017.2697942.

Author Response

Dear Reviewer,

Thank you for giving me the opportunity to submit a revised draft of our manuscript.  We have responded point by point to the comments in the attached pdf file.

Regards,

Farzaneh Bayati

Round 2

Reviewer 1 Report

I would like to thank the authors for improving the manuscript. Please see my remaining comments below.

Line 23. The format of referring to the references should be [1,2,3], not (1;2;3). Also [4,5,6] and [7,8]. Please check and correct.

Line 29. It is "Ghaderpour" not "Ghaderpoor".

Lines 39 and 45. It should be Wang et al. []. Please insert "et al." when there are more than two authors. When there are one or two authors, please write their full last names.For example, Line 56. Naghizadeh and Sacchi [26]. Or in line 89. Wu and Bai [36], etc.

Figure 5. Panel k. It looks like you took an screenshot. Please generate the figures more professionally with higher resolution.

Before Equation (11), you said "[34] proposed...". Please write it as "Nadakuditi and Optshrink [34] proposed...". Please check carefully the references to make sure you did not start a sentence with reference number.

Figure 6. Panels (d) and (f). The word "data" in the title of panels have missing letter "a".

Figure 11. Please generate a high quality figure with a larger font size.

Line 227-232. Please move this paragraph to the end of conclusions, i.e., there is no need to open a new section for Future Work.

Please carefully proofread the manuscript before re-submission.

Thank you

Author Response

Dear Dr. Ghaderpour,

I would like to thank you for revising our manuscript for the second time. The attached file is the answers to the comments.

Regards,

Farzaneh Bayati
